

# Robust active wake control in consideration of wind direction variability and uncertainty

Andreas Rott[1], Bart Doekemeijer[2], Janna Seifert[1], Jan-Willem van Wingerden[2], and Martin Kühn[1]

[1]ForWind - Center for Wind Energy Research, Institute of Physics, University of Oldenburg, 26129 Oldenburg, Germany
[2]Delft Center for Systems and Control, Delft University of Technology, 2628 CD Delft, The Netherlands

*Correspondence to:* Andreas Rott (andreas.rott@forwind.de)

**Abstract.** The prospects of active wake deflection control to mitigate wake-induced power losses in wind farms have been demonstrated by large eddy simulations, wind tunnel experiments and recent field tests. However, it has not yet been fully understood how the yaw control of wind farms should take into account the variability of current environmental conditions in the field and the uncertainty of their measurements. This research investigated the influence of dynamic wind direction changes on active wake deflection by intended yaw misalignment. For this purpose the wake model FLORIS was used together with wind direction measurements recorded at an onshore met mast in flat terrain. The analysis showed that active wake deflection has a high sensitivity towards short-term wind directional changes. This can lead to an increased yaw activity of the turbines. Fluctuations and uncertainties can cause the attempt to increase the power output to fail. Therefore a methodology to optimise the yaw control algorithm for active wake deflection was introduced, which considers dynamic wind direction changes and inaccuracies in the determination of the wind direction. The evaluation based on real wind direction time series confirmed that the robust control algorithm can be tailored to specific meteorological and wind farm conditions and that it can indeed achieve an overall power increase in realistic inflow conditions. Furthermore recommendations for the implementation are given which could combine the robust behaviour with reduced yaw activity.

## 1 Introduction

In recent years, more and more wind turbines have been installed in ever larger wind farms. On the one hand, this has advantages in logistics, network connection, maintenance, as well as the utilization of the limited suitable locations. On the other hand, this leads to situations where turbines are considerably more affected by harmful wake conditions. Downstream turbines are experiencing higher turbulence, which is generally associated with larger fatigue loads, and lower wind speeds, which results in a lower energy yield. In order to counteract this, different strategies are being investigated that try to reduce these negative wake effects. One approach to achieve this is active wake deflection by intended yaw misalignment, which was already investigated by (Medici and Dahlberg, 2003) in a wind tunnel experiment. While conventional, so-called greedy, turbine control seeks to optimize the operation of the individual turbines without taking into account the mutual effects on the other turbines, active wake deflection is the attempt to alter the trajectory of turbine wakes in order to improve the inflow conditions of downstream turbines. If two turbines are interacting through a wake, the deflection is achieved by deliberately introducing





a yaw misalignment of the rotor of the upstream turbine with respect to the wind direction. The rotor then generates a thrust force component that is perpendicular to the wind direction, which laterally deflects the wake. The goal is that the power gain of downstream turbines is higher than the power loss of the misaligned upstream turbine. More recent wind tunnel experiments (Bastankhah and Porté-Agel, 2015; Campagnolo et al., 2016) and large eddy simulations (LES) (Gebraad et al., 2014) demon-

strated the potential of such wake steering strategies for increasing the overall energy yield. Additionally, (Vollmer et al., 2016) investigated the influence of different atmospheric stabilities on active wake deflection in an LES study and analysed multiple sources that contribute to the uncertainty of the estimation of the deflected wake position. However, in both wind tunnel tests and conventional high-fidelity CFD simulations the inflow enters through a defined area, thus realistic dynamical changes in the wind direction are often not adequately reproduced. Fluctuating wind conditions and a high sensitivity towards the wind

direction make it difficult to take appropriate account of wakes in wind farm control in an uncontrolled environment like the free field.

The contribution of this article continues the investigations of (Seifert, 2015) and extends it by using methods of stochastic programming (Birge and Louveaux, 2011) to take better account of the uncertainties occurring in the field. This is done to derive a robust control strategy for the yawing of turbines, which is evaluated below. First results were published in the final report of

BMWi-funded research project CompactWind (Ahrens et al., 2016) and presented at the WindTech 2017 (Rott et al., 2017). The stochastic programming approach was also pursued in (Quick et al., 2017) for optimizing yaw angles for wake deflection while considering yaw errors only. In contrast this paper takes into account wind direction dynamics and measurement inaccuracy.

The objectives of this paper are 1) to analyse the impact of dynamical wind direction changes on active wake deflection strategies, 2) to introduce a methodology to optimize the yaw angle adjustment in a wind farm by taking these fluctuations and

measurement uncertainties into account 3) to propose open-loop control algorithms for active wake deflection in a wind farm.

## 2   Methods

In this paper, a quantitative analysis of wind direction variability and its effect on active wake deflection is carried out. Therefore, first the employed model for wake deflection (Section 2.1) and an analytical description of the statistics of wind direction variability (Section 2.2) are introduced. Based on this, an optimization of the yaw angle adjustments of all turbines in a wind

farm with respect to the relative change in the power output of the farm is established (Section 2.3) and evaluated for a fictitious reference wind farm (Section 2.4).

### 2.1   Wake deflection model

The investigation in this article is based on the FLOw Redirection and Induction in Steady-state (FLORIS) model (Gebraad et al., 2016), which has been specially developed for active wake deflection at the Delft University of Technology and the

National Renewable Energy Laboratory (https://github.com/wisdem/floris). A more elaborate description on control-oriented models in general, see (Boersma et al., 2017). The comparatively low computational complexity of the steady-state models makes it possible to perform optimization algorithms and validations on the basis of large datasets.





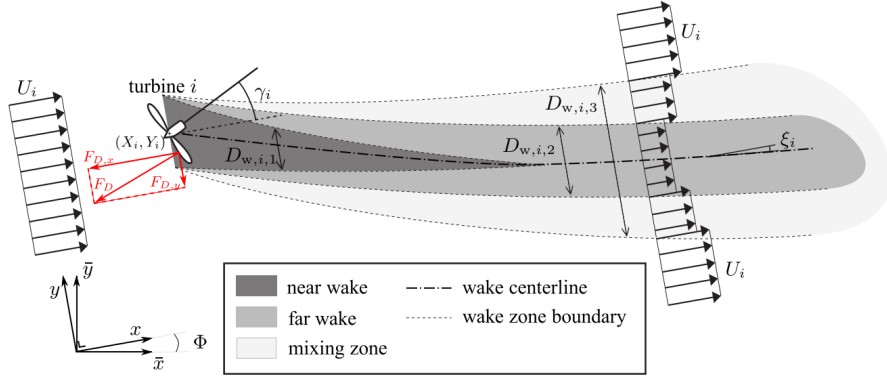

**Figure 1.** A schematic visualization of the wake model in FLORIS, containing three discrete wake zones. Furthermore, active wake deflection by purposely misaligning the turbine rotor with the flow is shown. (cf. Figure 4(a) in (Gebraad et al., 2016))

FLORIS extends the popular Jensen wake model (Jensen, 1983) to a more detailed wake description, containing three zones: a near wake zone, a far wake zone, and a mixing zone. See Figure 1.

These three wake zones each contain their own set of parameters for wake recovery and expansion, increasing the model's flexibility and fidelity. Furthermore, FLORIS uses the simplified analytical model from (Jiménez et al., 2009) which determines the wake deflection as a function of the turbine's thrust force and yaw angle. For multiple wake overlap situations, the "sum of squares" approach for the superposition of wake deficits is followed, first suggested by (Katic et al., 1986).

In short, FLORIS predicts the time-averaged steady-state conditions of the flow and each turbine's power capture as a function of each turbine's axial induction (i.e., a parametrization the generator torque and blade pitch angles), yaw angle, and atmospheric conditions inside a given farm. The applicability of the model has been demonstrated in high-fidelity simulations (e.g., (Gebraad et al., 2016)), wind tunnel tests (Schreiber et al., 2017), and even to some degree field tests (Fleming et al., 2017).

## 2.2 Statistical analysis of wind direction measurements

The wind direction angle can be expressed in radians $\varphi \in [0, 2\pi)$ as well as in degrees $\varphi \in [0°, 360°)$ interchangeably. Both representations are used in this article. Formulas are generally written using radians, while illustrations are presented in degrees using the standard conventions, i.e. 0° represents north and the rotation is clockwise.

The wind direction $\varphi \in [0, 2\pi)$ as input variable is a decisive parameter for the successful application of active wake deflection. In the field, however, the wind direction can change continuously and sometimes abruptly, as it can be seen in the example in Figure 2.

The turbulent changes in the wind direction are in contrast to the slowly reacting yaw mechanism of utility-scale wind turbines, which is usually driven by a wind direction measurement that is averaged over several minutes and is commonly controlled by a dead-band controller (Burton et al., 2011). Most of the time the yaw control is in standstill mode (Kim and Dalhoff,



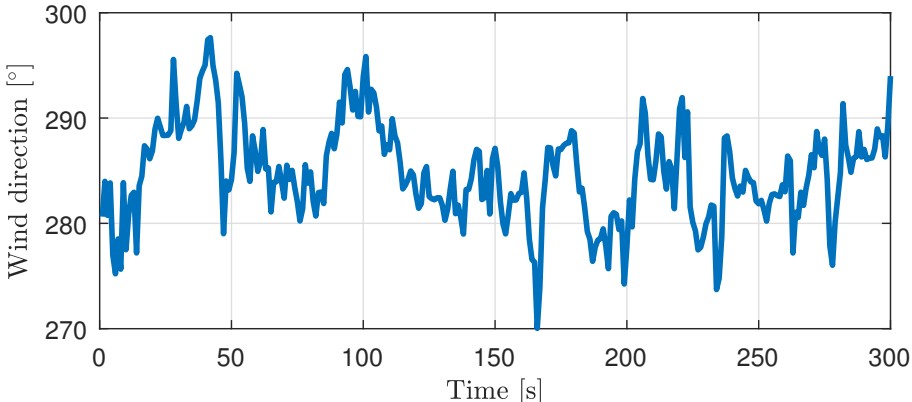

**Figure 2.** Exemplary 5 min time series of wind direction measurements recorded at an onshore test site in northern Germany sampled in 1 Hz resolution.

2014) to lower the yaw actuation, which means that the turbine maintains a constant yaw angle while it experiences varying wind directions. Although the details on the hysteresis of the yaw control depends on the manufacturer and is commonly kept confidential, in our experience the yaw angle remains constant for about 5 to 10 minutes in most cases before the yaw control corrects the yaw angle according to the changed wind direction. For this reason, we have studied the statistics of 5-minute

wind direction time series from 1 Hz measurement data denoted as $\Phi_t \in \{(\varphi_1, \ldots, \varphi_{300}) \in \mathbb{R}^{300} \,|\, \varphi_\tau \in [0, 2\pi), \tau = 1, \ldots, 300\}$, where the variable $t \in \mathbb{N}$ indexes successive time series. Figure 2 depicts an exemplary time series $\Phi_t$ from which a histogram is derived in Figure 3. A probability density function of a normal distribution (in red) with the same mean value (284.78°) and standard deviation (4.60°) as the measurement data is fitted to the histogram. Since a histogram depends very much on the binning, we added a quantile-quantile plot in Figure 3 (right), which is commonly used to compare distributions. Here, the mea-

surement data is compared with the fitted normal distribution. The measurement data is sorted by its values and plotted against the respective quantiles of the normal distribution. The better the distributions match, the more of the values lie on the straight red line. For the exemplary time series, both representations demonstrate the similarity to the normal distribution reasonably well, which agrees with the findings of (Gaumond et al., 2014), that wind direction behaviour is normally distributed within one averaging period, although Gaumond investigated 10 minutes time series. However, it should be noted that conventional

stochastic tools for the analysis of directional data are not generally valid, whereas circular statistics consider that e.g. $\varphi$ and $\varphi + k \cdot 2\pi$ for any $k \in \mathbb{Z}$ are identical angles on a standard circle. For example, the von Mises (or Tikhonov) distribution is typically used as an approximation of a wrapped normal distribution. Further, the directional mean $\varphi_{\mathrm{DM}} \in [0, 2\pi)$ differs from the arithmetic mean. It is defined as the angle of the sum of all unit vectors of the wind directions $\varphi_\tau, \tau \in \mathbb{N}$ (see Equation (1)), but in programming commonly the four quadrant arcustangens (atan2) operator is used for the computation:

$$\varphi_{\mathrm{DM}} := \arg\left(\sum_\tau \exp\left(\sqrt{-1} \cdot \varphi_\tau\right)\right) = \mathrm{atan2}\left(\sum_\tau \sin(\varphi_\tau), \sum_\tau \cos(\varphi_\tau)\right) \tag{1}$$





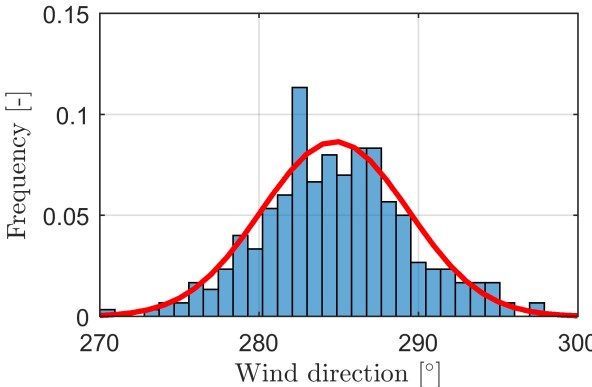
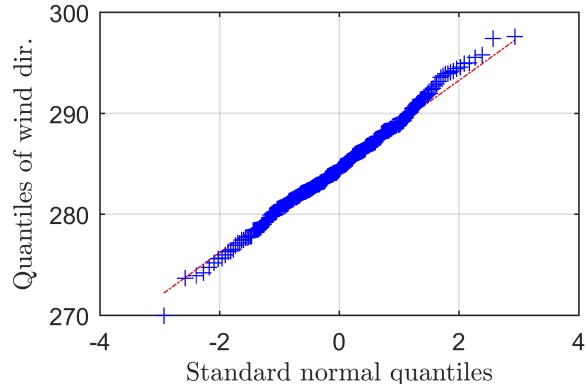

**Figure 3.** (left) Histogram of exemplary 5 min wind direction measurements, (right) quantile-quantile plot of the exemplary 5 min wind direction measurements

Nevertheless, since the wind direction distributions during a short time period include a relatively small sector compared to the whole circle, the differences between the normal distribution and a wrapped normal distribution are negligible. Therefore, and since it is comparatively less complex, we use the standard tools as mentioned above. But, since each angle can be expressed in multiple ways, we shift the transition angle at 360°/0° of each 5-min time series to its respective opposite angle of $\varphi_{\text{DM}}$. This can be achieved by the modulo operator (see Equation (2)), where $\varphi_{\text{raw}}$ refers to the original raw data.

$$\varphi := (\varphi_{\text{raw}} + \pi - \varphi_{\text{DM}} \bmod 2\pi) - \pi + \varphi_{\text{DM}} \tag{2}$$

In this way, we can apply the conventional calculation for the mean value and the standard deviation of the wind direction data $\varphi$. The validity of the assumption of an underlying normal distribution for 5 min wind direction data series is statistically tested in Section 3.1

## 2.3 Approach for optimization of yaw angles

In order to optimize the yaw settings of the turbines in a wind farm, we assume that the total power output of a wind farm consisting of $n$ wind turbines corresponds to the sum of the individual power outputs of the wind turbines. Conduction losses are therefore ignored. The turbines' power is estimated by FLORIS for a set of environmental conditions and control variables, i.e. the axial induction factor $a_j$ and the yaw angle $\gamma_j$ of each individual turbine $j = 1, \ldots, n$. With $\Gamma := \{\gamma_j \in [0, 2\pi) | j = 1, \ldots, n\}$ we denote the set of yaw angles of all turbines. Without loss of generality, we assume that the turbines run at a constant axial induction factor of $a_j = \frac{1}{3}$ for all $j$ and the power output $P_j$ of each turbine is normalized with respect to the power output of a turbine in undisturbed inflow conditions, since we are focussing on the influence of the yaw angle on the relative change of turbine power. Therefore, both $a_j$ and the wind speed are omitted in the following equations. $P_j$ depends on its own control variable $\gamma_j$, the wind direction as well as the yaw angles of all other turbines due to the aerodynamic interaction in the wind farm. Hence, we denote $P_j = P_j(\gamma_1, \ldots, \gamma_n, \varphi)$.





Next we are introducing two different optimizations of the yaw angles which differ in the description of the wind direction variability. Firstly, we are neglecting wind direction changes within the investigated time period. The optimization problem formulated in Equation (3) aims at finding the set of optimal yaw angles $\Gamma^{\mathrm{opt}}(\varphi) = \{\gamma_1^{\mathrm{opt}}(\varphi), \ldots, \gamma_n^{\mathrm{opt}}(\varphi)\}$, which maximizes the power output of the wind farm for the prescribed wind direction $\varphi$. We will refer to this as the conventional optimization in the following.

$$
\begin{aligned}
&\text{find}\\
\Gamma^{\mathrm{opt}}(\varphi) = \underset{\Gamma}{\mathrm{argmax}} &\sum_{j=1}^{n} P_j\left(\gamma_1, \ldots, \gamma_n, \varphi\right)\\
&\text{for the wind direction } \varphi
\end{aligned}
\tag{3}
$$

Secondly, the optimization of the yaw angles is formulated more robust towards wind direction dynamics and uncertainties in the measurements. Instead of considering only one wind direction, the new problem description aims at finding the set of optimal yaw angles for a distribution of wind directions weighting the results for every individual wind direction by its probability of occurrence. This is achieved in Equation (4) by a probability density function $\rho(\varphi)$, which represents wind directional variation and uncertainties stochastically. We will refer to this approach as the robust optimization in the following.

$$
\begin{aligned}
&\text{find}\\
\Gamma^{\mathrm{opt}}(\rho(\varphi)) = \underset{\Gamma}{\mathrm{argmax}} &\int_0^{2\pi} \rho(\varphi) \sum_{j=1}^{n} P_j\left(\gamma_1, \ldots, \gamma_n, \varphi\right) \mathrm{d}\varphi\\
&\text{for a probability density function } \rho(\varphi).
\end{aligned}
\tag{4}
$$

This formulation is a generalization of the conventional optimization, which is obtained when we insert the dirac delta function for the probability density function.

To solve the integral in Equation (4) the cost function is discretized. For the calculation we have chosen a step size of $1°$, which corresponds reasonably well with the measuring accuracy of wind vanes and still gives a good representation of the distribution.

A number of algorithms can be used for the computation of these kinds of optimization problems, including the intuitive game theoretic approach presented in (Marden et al., 2012). This algorithm has the benefit that it works on complex, non-linear problems and does not depend on any gradients, but unfortunately it converges relatively slow. For this research therefore, we used the pattern search algorithm (Audet and Dennis, 2002), which has similar properties but we experienced a faster convergence speed.

## 2.4 Test case

In order to evaluate the control strategies derived from the conventional and the robust optimization, a case study is performed for a reference test wind farm. It consists of 9 NREL 5-MW turbines (Jonkman et al., 2009) with a rotor diameter of $126\,\mathrm{m}$ in a grid layout (see Figure 4). The turbines are situated relatively close to each other, so that strong wake effects occur, as the





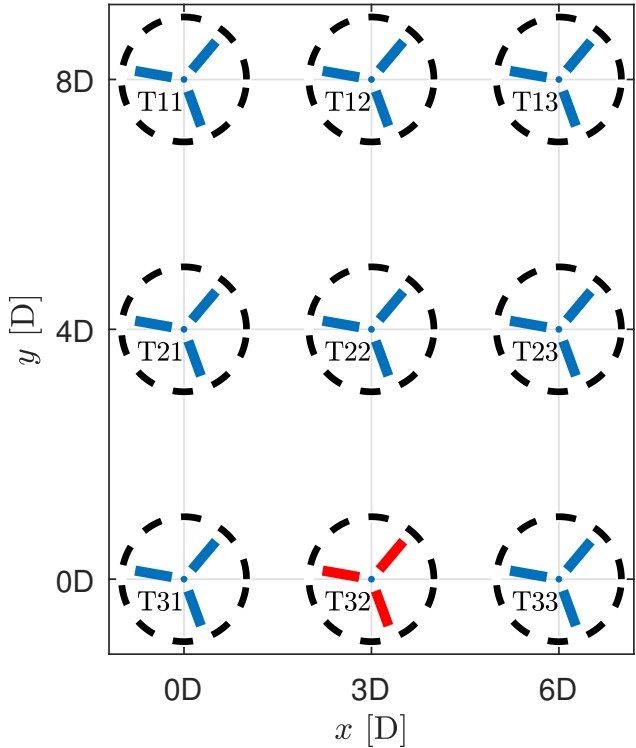

**Figure 4.** Layout of the reference wind farm. The reference turbine (T32) is marked in red.

active wake deflection is of special interest for such situations. With this layout the distances between adjacent turbines are
3D horizontally, 4D vertically and 5D diagonally. These values are comparable to the dense spacing in the offshore wind farm
Lillgrund (Papatheou et al., 2015).

5  The focus of the investigation is to determine the sensitivity of the control strategies with regard to wind direction variability
and uncertainty. In order to perform this investigation with realistic data, wind direction measurements from a met-mast in
$92\,\mathrm{m}$ height at a near coastal test site in Brusow, north-eastern Germany, were used as input. The surrounding area was mostly
flat, some complexity was added by a nearby forest. For more details we refer to (Bromm et al., 2018). Of the available
measurements, only data with a 5 minutes average wind speed between $3.5\,\mathrm{m/s}$ and $14\,\mathrm{m/s}$ was used, since power optimization
is only of interest when the turbines operate above cut-in and below rated wind speed. This corresponds to $87\,\%$ of the data
10  collected between June 30th and November 22nd of 2016, which gives a total of $N = 35,586$ of 5 minute time series. Figure
5 illustrates a wind rose of the data showing the frequencies of occurrence of the directions.





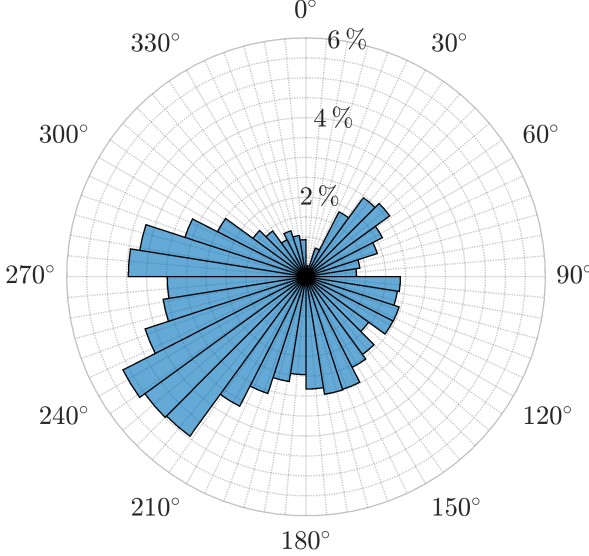

**Figure 5.** Wind rose of 1 Hz measurement data recorded at a met mast in Brusow, north-eastern Germany

## 3 Results

In this section, the method for investigating the statistical properties of wind direction changes using measurement data is examined (Section 3.1). In Section 3.2 the two methods for yaw angle optimization were applied to the reference wind farm and in Section 3.3 open-loop control algorithms are derived from the optimizations and are evaluated using the measurement

5 data.

### 3.1 Stochastic properties of wind direction measurements

As mentioned in Section 2.2, we analysed the stochastic properties of 5 minute time series of 1 Hz wind direction measurements denoted as $\Phi_t$. Specifically, we want to verify the hypothesis that $\Phi_t$ can be approximated statistically by a normal distribution as indicated by Figure 3. Therefore, we performed a Kolmogorov-Smirnov test on the subject (Chakravarti and Laha, 1967).

10 In this fitting test, the empirical distribution of $\Phi_t$ is compared to a normal distribution and a critical value is calculated. This value, together with the chosen significance level, determines whether to accept or reject the hypothesis. In our case, 70.58 % of the measurements used for this investigation passed the test for a significance level of 5 %. From this we draw the conclusion that in most cases 5 minute wind direction time series can be reasonably represented by normal distributions. For normally distributed data, the mean value and the standard deviation are sufficient to describe the distribution completely, so the standard

15 deviation is of particular importance for our measurements. For 95 % of the used data the standard deviation was between 0.67° and 12.67°, with an average of 5.26°.





However, the dynamics of the wind direction is not the only uncertainty factor in rotor alignment. In addition, there are inaccuracies in the determination of the wind direction and the alignment of the turbine. Such types of measurement errors are commonly assumed to be independent and normally distributed (Murcia et al., 2015). For a random variable that is based on two or more independent distributions, its distribution can be determined by the convolution of the individual distributions. In the case of normal distributions, the convolution results again in a normal distribution, in which the variances of the underlying distributions are added arithmetically.

For these reasons, a normal distribution is chosen for the probability density function of the measured wind direction in the robust optimization, which represents the assumed uncertainty and variability of the wind direction. Since the support of the normal distribution is unrestricted, we limit the range to $\varphi \pm 2\sigma$, in order to cover the majority ($\approx 95.45\,\%$) of the occurring events. The advantage of using a normal distribution is that the robustness of the optimization is governed by only one variable, i. e. the chosen standard deviation $\sigma$ of the distribution, which we will refer to as the robustness parameter. For completeness it should be mentioned that commonly the wind direction measurement by nacelle anemometry is subject to a bias as well. Such an offset has a clearly degrading effect on active yaw deflection control and should be reduced by proper calibration of the wind vane. For this purpose several practical procedures are available as for instance demonstrated by (Mittelmeier et al., 2017; Mittelmeier and Kühn, 2018). Without loss of generality we assume in our analysis that a wind direction bias is negligible.

### 3.2 Results of the yaw angle optimizations

The solutions of the conventional and the robust optimization respectively, are the optimal yaw angles of all the turbines for all wind directions. In Figure 6 the results of the yaw schedule of only one turbine (T32) is displayed for four cases. In the following, we will refer to this turbine as the reference turbine. It is located in the centre of the southernmost row of the wind farm and it is highlighted in red in Figure 4. Different robustness parameters $\sigma = 0°, 4°, 8°$ were chosen and the results are compared to the baseline yaw schedule, which is the case when there is no intentional yaw misalignment, represented by the grey diagonal line. The black graph refers to the conventional optimization corresponding to $\sigma = 0°$. The blue and red graph demonstrate the results of the robust optimization with $\sigma = 4°, \sigma = 8°$, respectively.

In Figure 6 it can be seen that the deviations from the baseline become smaller with increasing robustness parameter. In the black graph ($\sigma = 0°$), the yaw angles have relatively large deviations in the wind sector from roughly 70° to 290°. In seven distinct situations the yaw angle rapidly changes from a positive to a negative misalignment, which means that the yaw angle rotates contrary to the wind direction. This increases the overall yawing activity of the turbine and it is generally undesirable and should therefore be used with caution. These situations correspond to the angles at which at least one turbine is in the full wake of the reference turbine. In the remaining wind direction sector (290° to 70°) there is no yaw misalignment, since in these situations the wake of the reference turbine does not affect the other turbines in the wind farm.

In the blue graph ($\sigma = 4°$), the yaw misalignment is considerably reduced compared to the first case. The number of fluctuations, where the yaw misalignment rapidly changes from positive to negative, is reduced to five. This correspond to the angles at which one of the direct neighbouring turbines is in the full wake of the reference turbine.





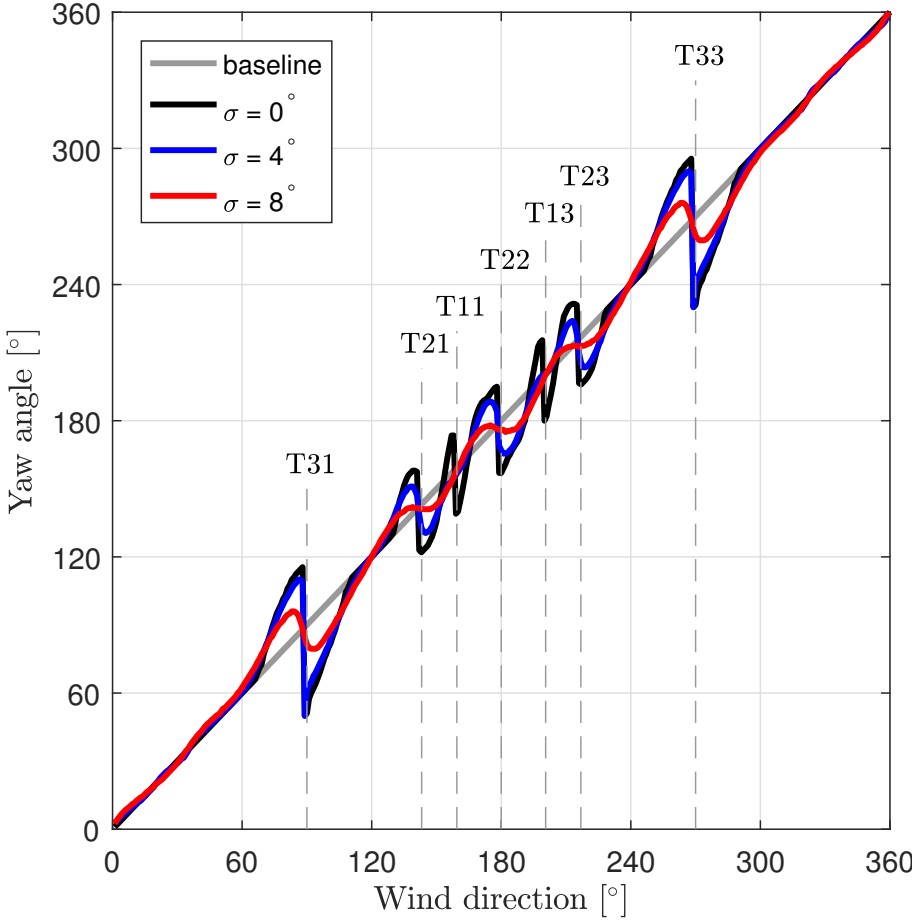

**Figure 6.** Optimized yaw angles of the centre turbine in the southernmost row (T32) for three different robustness parameters $\sigma = 0°, 4°, 8°$. In addition, directions are marked and named accordingly, at which neighbouring turbines are located downstream.

In the red graph ($\sigma = 8°$), the deviations from the baseline decrease further. Only at angles around $90°$ and $270°$, for which the turbines affected by the wake of the reference turbine are the closest, the yaw angle visibly rotates contrary to the wind direction. The other fluctuations are smoothed out to plateaus-like segments. At these plateaus the reference turbine maintains a nearly constant yaw angle for a wind sector of about $\pm7°$ to $\pm9°$ around the direction of maximum interaction. In these special

5  situations, the orientation of the reference turbine points almost exactly to the adjacent downstream turbine. This ensures that the wake is deflected to the correct direction (away from the downstream turbine) without adjusting the yaw angle, even if the wind direction changes within the specified sector. We refer to this special case hereinafter as passive wake deflection. It will be further discussed in Section 4.

We elaborate an exemplary case to better understand the impact of wind directional variation and uncertainty on active

10  wake deflection for the different robustness parameters. For this purpose, we select an arbitrary wind direction that is assumed




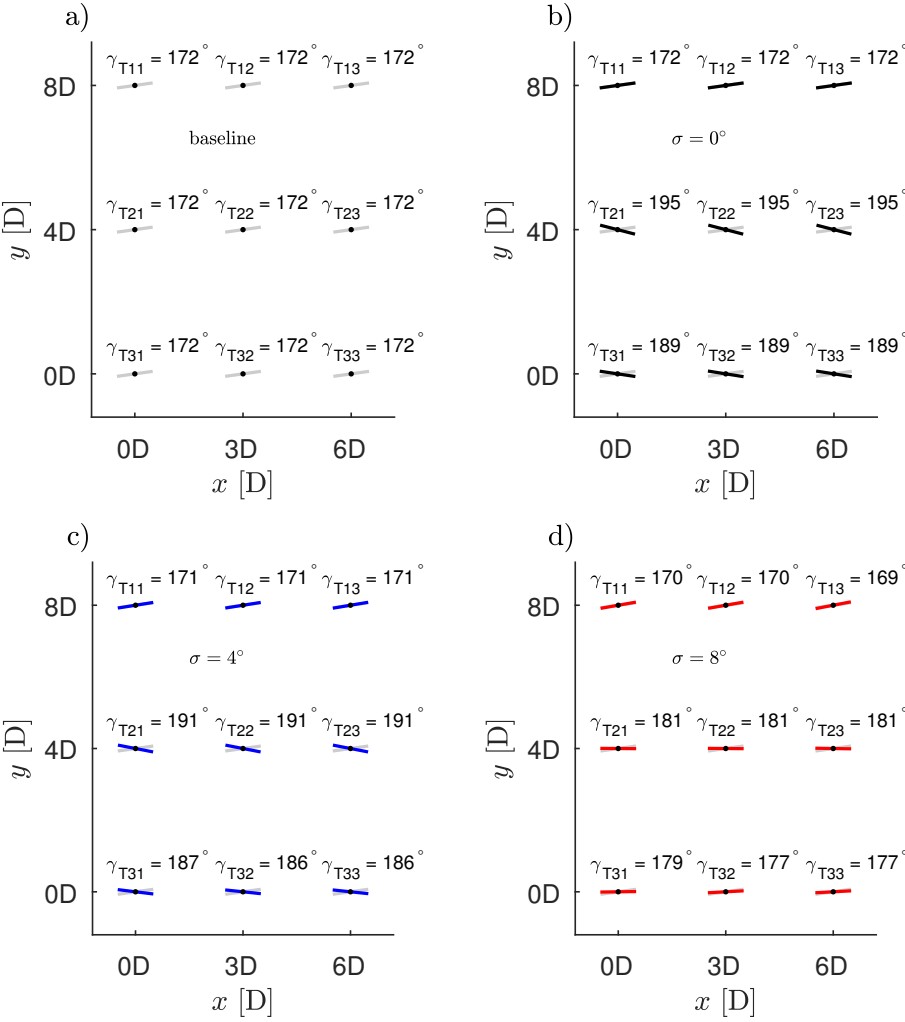

**Figure 7.** Illustration of the yaw angles of the reference wind farm for an estimated wind direction $\varphi_{\mathrm{est}} = 172°$ according to the different optimizations.

to predominate at the moment and to which the turbines adjust according to the respective optimization. We call this wind direction the estimated wind direction $\varphi_{\mathrm{est}}$. Then we analyse how performance of the different optimized yaw settings changes when the actual wind direction deviates from the estimated wind direction. As an illustrative example we choose $\varphi_{\mathrm{est}} = 172°$, which is a situation where strong wake effects occur. The optimized yaw angles of the turbines are determined by the robust

5    optimization for the robustness parameters $\sigma = 0°, 4°, 8°$, respectively. The resulting yaw angles of all turbines in the reference wind farm for this particular case are displayed in Figure 7. For a better comparison of the optimized yaw angles to the baseline, the yaw angles according to the baseline are depicted in each of the four illustrations in Figure 7 by a grey line.





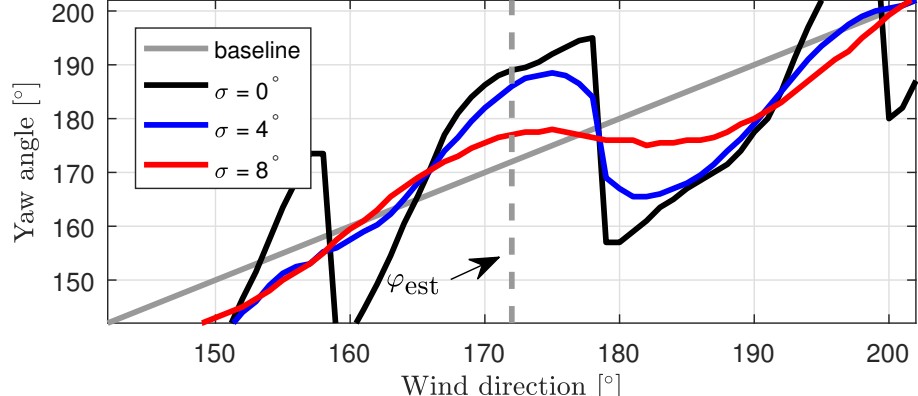

**Figure 8.** Optimized yaw angles of the reference wind turbine for $\varphi_{\text{est}}$.

Again, it can be seen that the deviation becomes smaller with increasing robustness parameters. However, the results of the robust optimization with $\sigma = 4°$ and $\sigma = 8°$ appear unexpected at first in regard to two aspects.

Firstly, the yaw angles of the northernmost turbines (T11, T12, T13) slightly deviate from the inflow direction, although there are no downstream turbines to consider. The observed small positive offset is opposite to the negative yaw misalignment

of the other turbines. The reason for this is that the robust optimization considers all inflow directions around the estimated wind direction in its objective function. The turbines in the northernmost row align themselves towards inflow directions from where less wake effects occur, so they can produce more power in these situations. Consequently, the power output is reduced for inflow direction in the opposite direction. However, the relative power loss is smaller, since in this case the turbines already produces less power due to the stronger wake effects.

Secondly, the optimized yaw angle of the T31 differs slightly from the optimized yaw angle of the turbines T32 and T33. This is due to the location of this turbine at the edge of the wind farm. If the wind would turn counter-clockwise, the wakes of the turbines T32 and T33 affect the rest of the wind farm, e.g. T12. This is not the case with T31, so this turbine does not have to take this into account and applies a larger yaw misalignment.

In order to further analyse the results of the optimizations we are looking in detail to the reference turbine T32. The set-
points for the yaw angle of the reference turbine (T32) is displayed in Figure 8 which is an enlarged part of Figure 6 for angles around $\varphi_{\text{est}}$. The yaw angle of the reference turbine according to the conventional optimization ($\sigma = 0°$) is $\gamma = 189°$, which equals an intentional misalignment of $17°$. The robust optimization with $\sigma = 4°$ results in a yaw angle $\gamma = 186°$ and the robust optimization with $\sigma = 8°$ gives a yaw angle $\gamma = 177°$.

For these settings, we can now calculate the power output of the reference wind farm for different wind directions with the
help of FLORIS. The normalized power difference $P_{\text{diff}}(\varphi) := \frac{P_{\text{opt}}(\varphi) - P_{\text{baseline}}(\varphi)}{\max_\varphi(P_{\text{baseline}}(\varphi))}$ for the robustness parameters $\sigma = 0°, 4°, 8°$ are displayed in Figure 9. The graph illustrates the power gain of the robust optimizations and how it is affected if the wind direction $\varphi$ deviates from $\varphi_{\text{est}}$. All three yaw settings achieve power increase close to $\varphi_{\text{est}}$, which is the design point for the optimization in this case. As expected, at the design point the conventional optimization ($\sigma = 0°$) has the highest gain of the





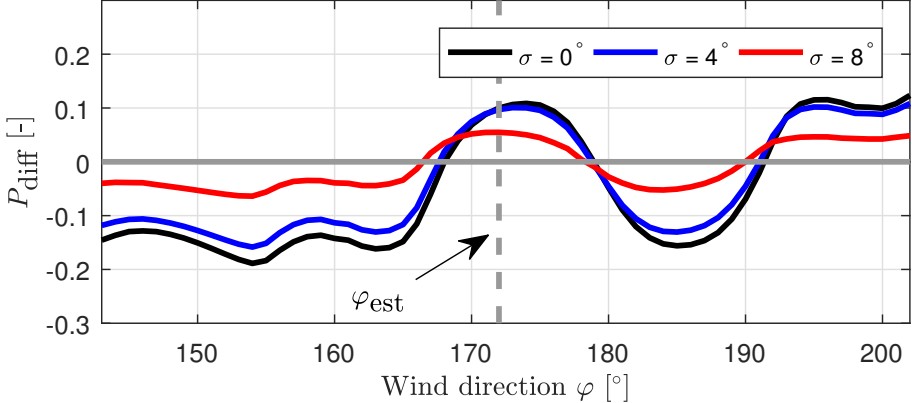

**Figure 9.** Normalized power gain of the conventional optimization (black) and of the robust optimization (blue and red) compared to the baseline (grey) for $\varphi_{\text{est}} = 172°$

optimizations, but it also has the largest drop further away from $\varphi_{\text{est}}$. While the robust optimization with $\sigma = 4°$ is only slightly below the conventional optimization at the design point, it performs better in the outer regions (e.g. around $160°$ or $185°$). For the robust optimization with $\sigma = 8°$ the maximum gain is lower, but the losses in the outer regions (e.g. around $160°$ or $185°$) are strongly reduced.

For wind direction above roughly $191°$ the power difference $P_{\text{diff}}$ becomes positive again for all robustness parameters. This is due to the fact, that the turbines that follow the baseline ($\gamma_j = \varphi_{\text{est}}$ for $j = 1, \dots, n$) get large yaw misalignments and thus the power output $P_{\text{baseline}}$ is significantly reduced. In summary, the graphs show that for all three cases a deviation of the wind direction from the estimated wind direction can lead to significant power losses, rather than power increases.

### 3.3 Evaluation of control algorithms

In this section, the following four time-dependent yaw control algorithms are introduced and evaluated with the help of FLORIS on the basis of wind direction measurements.

1. greedy yaw control: This reference control is derived from the baseline, it refers to the situation, that every individual turbines tries to locally maximize its power output by yawing directly into the wind direction without any intentional yaw misalignment. The term greedy control was introduced by Marden et al in the scope of induction-based wind farm control (Marden et al., 2012).

2. conventional wake deflection: This active wake deflection control scheme applies the yaw angles calculated by the conventional optimization ($\sigma = 0°$). For a given $\varphi_{\text{est}}$ the yaw control algorithm uses the precalculated yaw angle of the conventional optimization in the form of a lookup table.




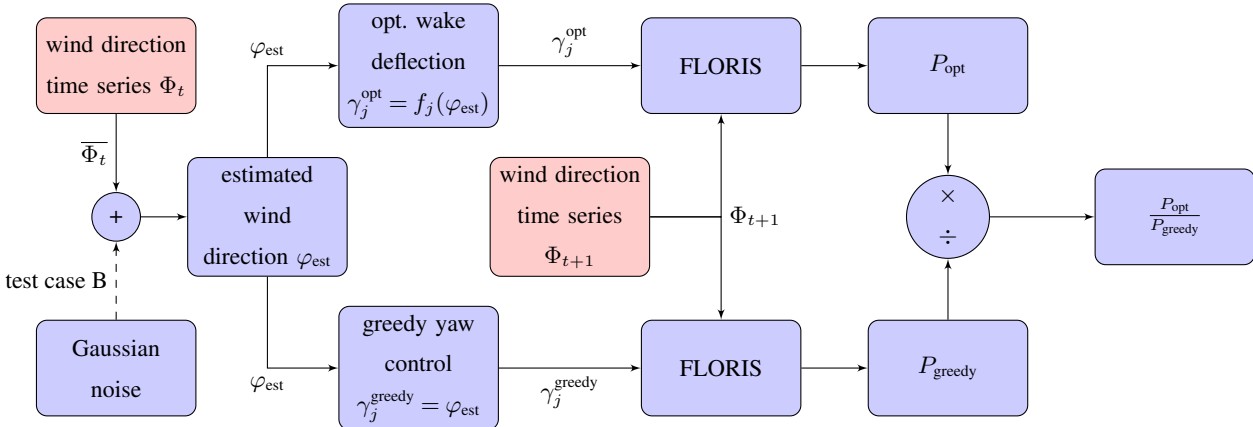

**Figure 10.** Evaluation process of the control schemes with wind direction measurements. Red boxes mark the steps in which the measured data is used as input.

3. robust wake deflection ($\sigma = 4°$): The open-loop control of the robust wake deflection works in the same way as the conventional wake deflection, but this time the yaw angle settings of the turbines from the robust optimization with $\sigma = 4°$ are used.

4. robust wake deflection ($\sigma = 8°$): Analogous to the case before, but $\sigma = 8°$.

In the next step we are extending our evaluation on the above-mentioned four control strategies based on the actual time series from the wind direction measurements. Two test cases A and B are each analysed for three different robustness parameters ($\sigma = 0°, 4°, 8°$). The evaluation process and the test cases are described in the following and illustrated in Figure 10.

First, the 1 Hz wind direction data $\varphi$ was split up in 5 minute time series $\Phi_t \in \mathbb{R}^{300}$, $t = 1, \ldots, N$, with $N = 35,586$, in the same manner as it was done in Section 2.2. For test case A the estimated wind direction is defined as the mean wind

direction $\varphi_{est} := \overline{\Phi_t}$, indicated by the $\overline{\cdot}$-operator, and passed as input to the open-loop control schemes. For test case B, an additional Gaussian error $\theta$ with a standard deviation of $4°$ is added to the mean wind direction $\varphi_{est} := \overline{\Phi_t} + \theta$, to simulate additional inaccuracies like for example measurement uncertainties, yaw hysteresis and alignment errors. The output of the optimized wake deflection are the optimized yaw angles $\gamma_j^{opt}(\varphi_{est}) = f_j(\varphi_{est})$ of all turbines $j = 1, \ldots, n$. The function $f_j$ represents the yaw schedule of the $j$'s turbine according to Section 3.2. To evaluate the success of the control strategy we

compare it to the baseline, which is the greedy yaw control. Therefore, the estimated wind direction is also passed to the greedy yaw control scheme $\gamma_j^{greedy}(\varphi_{est}) = \varphi_{est}$ for all $j$. Next, the output of both control schemes $\gamma_j$ and the next time series of wind direction $\Phi_{t+1}$ are passed to FLORIS and the average power outputs $P_{opt}$ and $P_{greedy}$ for the 5 minutes of wind direction data are computed for the optimized and the greedy yaw control. This step simulates, that the yaw mechanism of the turbines does not constantly correct the yaw angle and can only react retroactively to changes in the wind direction. Finally, the power output

for the optimized wake deflection $P_{opt}$ is compared to the power output of the greedy yaw control $P_{greedy}$ and the results $\frac{P_{opt}}{P_{greedy}}$ are displayed in Figure 11 and Figure 12 for test case A and B, respectively.



Figure 11 and Figure 12 illustrate the relative power change ($y$-axis) over the estimated wind direction ($x$-axis) in a scatter plot, where one dot represents one 5 minute time series $\Phi_t$ each. As mentioned before, this evaluation has been carried out for the three different active wake deflection controls with robustness parameters of $\sigma = 0°, 4°, 8°$. Therefore, both figures consist of three graphs each.

Starting with test case A, the upper graph of Figure 11 displays the result of the conventional wake deflection. It can be seen that the relative power change is highly scattered around the neutral value of 1. This happens mainly for wind directions with strong wake effects. In these cases, both relatively large power gains and losses occur. A surplus in performance is achieved if the wake deflection works as intended, indicated by a value above 1. A value below 1 means a power loss, which arises when the fluctuations in the wind direction or inaccuracy in its determination are too large. Overall, an average relative performance

gain of $0.6\,\%$ was achieved in this test case.

In the middle graph the result of the robust wake deflection for $\sigma = 4°$ is presented in blue, while the result of the upper graph is displayed in black in the background. In the comparison one can see that the spread decreases slightly. Fewer extreme cases occur, both in terms of performance increase and losses. Moreover, the centre of the distribution is shifted towards a power increase resulting in a higher average relative performance gain of $1.44\,\%$. Similar to before, the lower graph illustrates

the difference between the robust wake deflection with $\sigma = 8°$ (in red) to the conventional wake deflection (in black). The scattering of the values decreases further here, while the average relative performance gain is $1.39\,\%$.

Figure 12 depicts the results of test case B in the same manner as before. An additional Gaussian error was added to the estimated wind direction in this case simulating measurement noise. As a consequence, lower values are achieved on average for each wake deflection approach. Although this is hard to see in the upper graph, the average relative performance gain is

reduced to $-0.49\,\%$. This example proves that the active wake deflection can fail its objective on average if uncertainties are not taken properly into account in the control strategy.

The introduction of additional uncertainties also affects the robust wake deflection, but the effects are not as strong as with the conventional wake deflection. The robust wake deflection with $\sigma = 4°$, presented in the middle graph still reaches a positive average relative performance gain of $0.61\,\%$. As expected the robust wake deflection with the highest robustness parameter

($\sigma = 8°$) proves to be less affected by the given additional uncertainties. In the lower graph results have the smallest change in performance and achieves the best average value of $1.05\,\%$ in this scenario.

## 4 Discussion

The evaluation of the yaw angle optimization and of the associated yaw control algorithms are based on real dynamic wind direction measurements, but for the calculation of the wake losses and the power output, a simplified steady-state wake model

is used, which approximates the average wake flow. In addition, we have limited our investigation to the partial load range, which we consider to be the most important. In this case, we assumed a constant thrust coefficient of the turbines for the analysis. Hence the work here is intended to serve as proof of concept and as the basis for further investigations. Following this




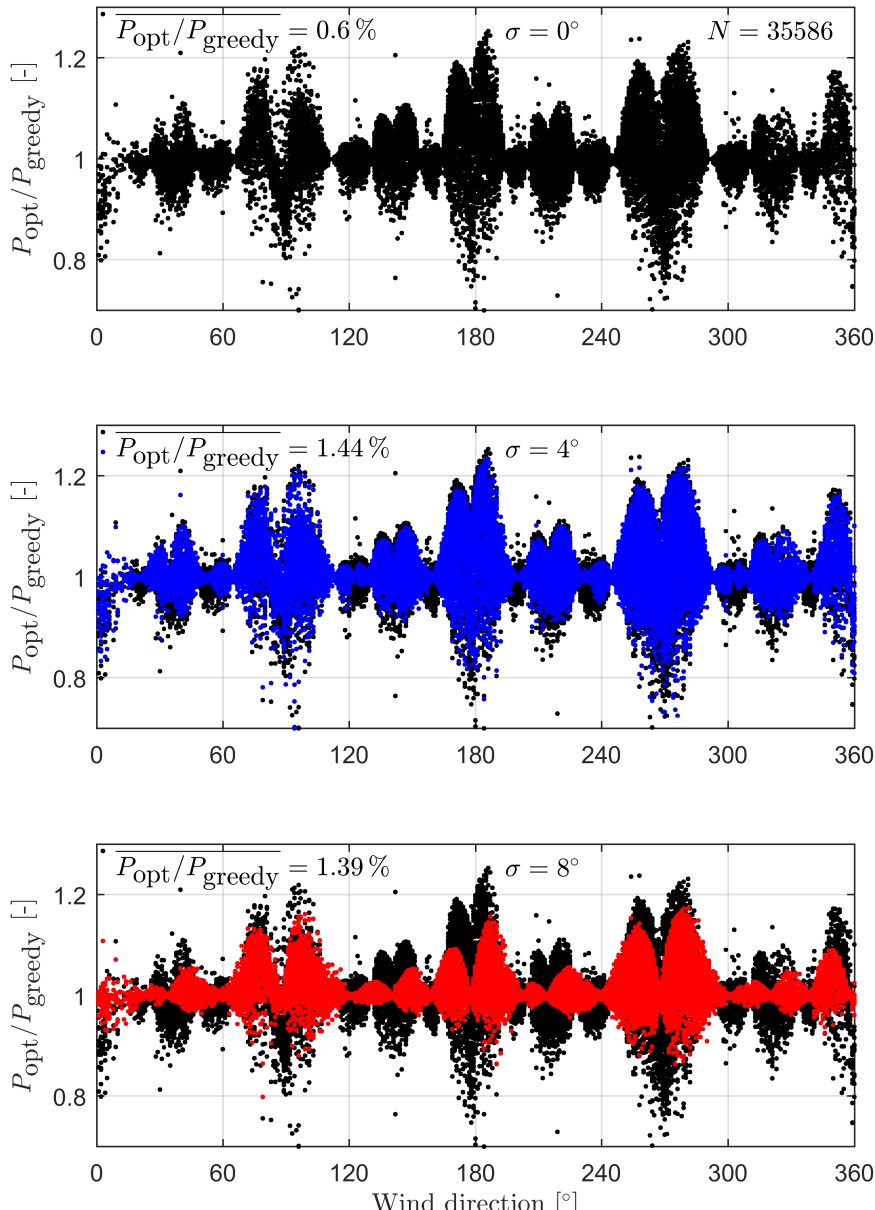

**Figure 11.** Test case A: Evaluation of the relative power gain over the estimated wind direction $\varphi_{est}$ from the measured time series of 5 minutes averaged wind direction during a duration of approx. 5 months. a) conventional wake deflection (black), b) and c) robust wake deflection for $\sigma = 4°$ (blue) and $\sigma = 8°$ (red), respectively




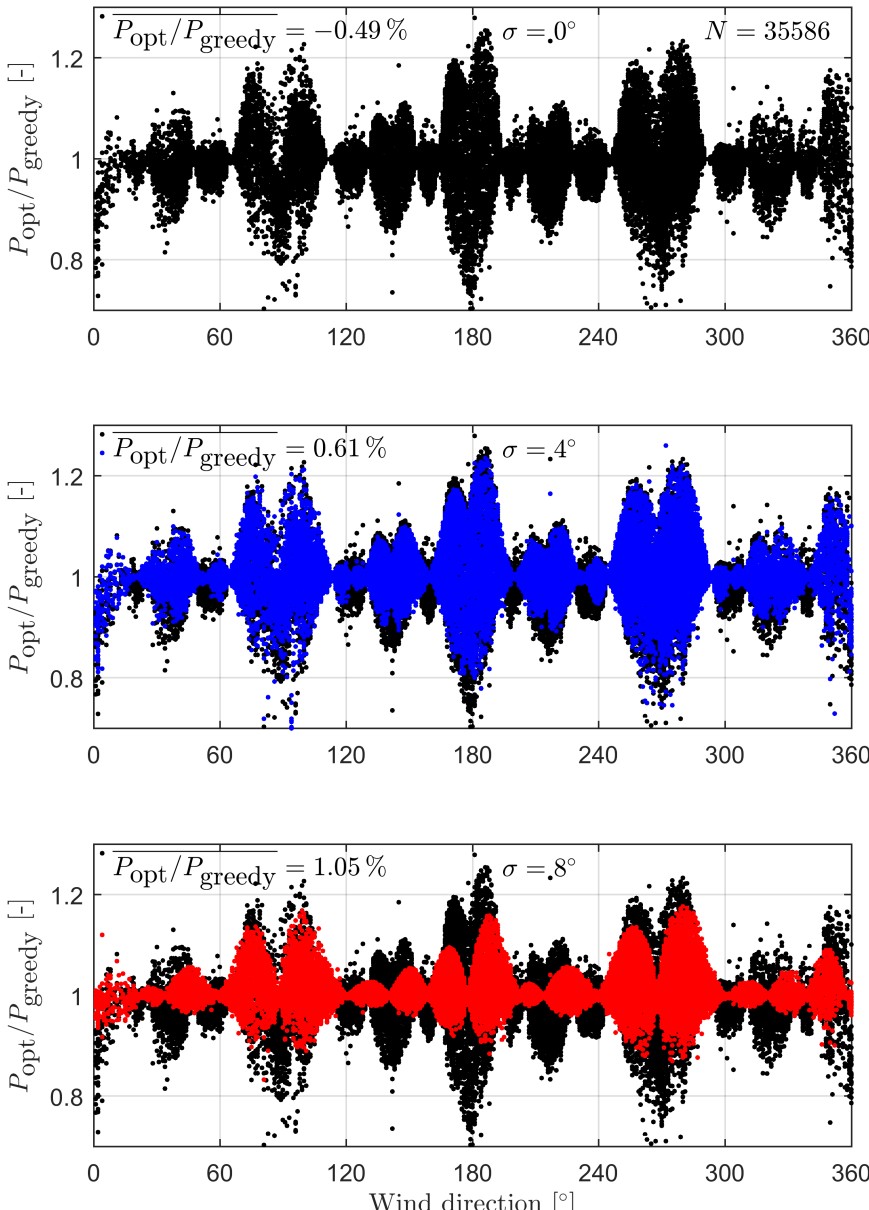

**Figure 12.** Test case B: Evaluation of the relative power gain over the estimated wind direction $\varphi_{\text{est}}$ from the measured time series of 5 minutes averaged wind direction during a duration of approx. 5 months. a) conventional wake deflection (black), b) and c) robust wake deflection for $\sigma = 4°$ (blue) and $\sigma = 8°$ (red), respectively





research, qualitative investigations based on large eddy simulations and free field experiments are proposed. When large eddy simulations are used one should take care that they properly reproduce real wind direction dynamics.

The relative power gain used here as a key performance indication must not be confused with a pure increase in the annual energy production (AEP). For a reliable estimation of the AEP a time series of both wind speed and direction of an average entire year together with the wind turbine power curve and availability as well the wind farm layout has to be available. The direct use of the commonly applied Weibull distribution of the mean wind speed would be insufficient. Since in this paper we wanted to focus on comparing the efficiency of the different control strategies in the partial load range, we used the relative power increase instead of the AEP.

In this study, the robustness parameter was deliberately set to a fixed value for the entire evaluation period in order to demonstrate its influence and effects. A meaningful refinement of the algorithm would be to utilize a variable robustness parameter and adapt it to the ambient conditions, e.g. the mean wind speed, turbulence intensity and atmospheric stability. Observations and large eddy simulations (Vollmer et al., 2016) indicate that with a stable stratification and associated low turbulence intensity considerably lower wind direction changes occur.

The presented results are potentially of significant importance for implementing active wake deflection in the field. The simplicity of the presented open-loop robust control algorithm makes it easy to integrate it into a real yaw control system, which offers the possibility to obtain further insights with the assistance of field campaigns. For this purpose, the wind farm layout and the turbine characteristics must be known for the calculations with the wake model, in addition the global alignment of the turbines should be as correct as possible and the wind measurements must be relatively reliable. If these requirements are met, the optimized yaw schedules can be calculated for the individual turbines and the robust wake deflection can be used. In principle, the robust wake deflection is even a decentralized control system, since each turbine follows its own optimized yaw set points independently of the others. However, in practice a wind farm regularly undergoes topology changes. This means that turbines change their status and are switched off if necessary. In such a case, the optimized wake deflection of at least the adjacent turbines should be deactivated for the corresponding wind direction sector and the greedy control should be used. A straightforward adjustment would be, for example, the switch to the greedy control for these turbines in the respective wind sector.

Given the industry's interest in easy and robust solutions a particular implementation could be the so-called passive wake deflection. This means that the yaw angle of an upstream turbine is set to a constant value for certain wind direction sectors. 'Passive' refers in this context to the strongly reduced yaw activity in comparison to the large yaw amplitude in the case of the conventional yaw angle optimization discussed in Section 3.2. This idea is derived from the example in Figure 8 and is further illustrated in Figure 13. In this case, according to the robust optimization with $\sigma = 8°$ the yaw angle of the reference turbine for a wind direction sector from $\varphi_{\mathrm{est}} = 170°$ to $\varphi_{\mathrm{est}} = 188°$ remains between $\gamma = 175°$ to $\gamma = 178°$. The adjustment of the control would be to fix the yaw angle for such sectors to a constant value, e.g. $\gamma = 176.5°$. Then within such a sector, the orientation of the turbine would be directed almost exactly to next neighbouring downstream turbine. The wake deflection is not particularly strong in such a case, but the advantage is, that the wake for all wind directions within this sector is automatically deflected away from the downstream turbine, making the application very reliable. Another benefit of such an implementation of the





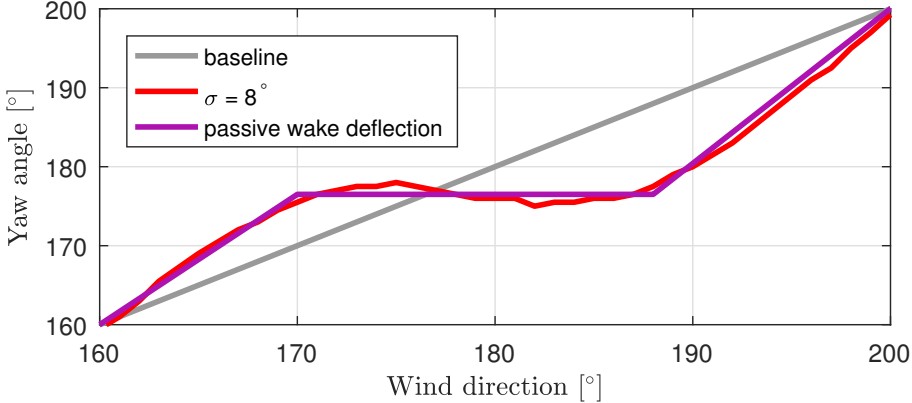

**Figure 13.** Exemplary illustration of the yaw angle set points of the reference turbine T32 according to the passive wake deflection.

robust control would be the possibility to reduce the yawing activities of the turbine significantly by keeping the intended yaw misalignments relatively small, if integrated correctly into the existing yaw control. This is in contrast to the increased yaw activity, which is associated with the conventional wake deflection control ($\sigma = 0°$) (see Figure 6.

## 5   Conclusions

The aim of this research was to demonstrate the influence of dynamic wind direction changes on active wake deflection and to present its potential to increase the energy yield of a wind farm in a realistic environment, if wind direction dynamics and associated measurement uncertainties are considered properly. Therefore, we first examined the stochastic properties of wind direction measurements and confirmed that a normal distribution is a useful approximation. Next, we demonstrated that the high sensitivity towards wind direction changes poses a risk for the successful application of active wake deflection. To cope

with these fluctuations and uncertainties, a robust optimization approach for the yaw angles of all wind turbines in a wind farm as function of the wind direction and the wind direction variability was introduced. The method takes dynamic wind direction changes and inaccuracies in the determination of the wind direction into account within a statistical framework in the optimization.

The results indicate that, in an evaluation of different open-loop contgrol algorithms with real wind direction time series, the

robust optimization can successfully increase the performance of a reference wind farm, while the conventional optimization neglecting wind directional dynamics and uncertainties can lead to a decrease in power output compared to greedy control without any attempt of wake steering.

The introduced robustness parameter $\sigma$, i.e. the standard deviation of the combined normal distribution of the wind direction and the associated measurement uncertainty, is a useful quantity to tailor the robust control to specific meteorological and

wind farm influences. If a larger robustness parameter, similar to the analyse example of $\sigma = 8°$, is appropriate the yaw control algorithm might be simplified to the so-called, passive wake deflection. In such an approach the yaw angle is kept constant in a





certain angular sector which significantly eases the implementation and reduces the required yaw activity to achieve favourable wake steering.

*Acknowledgements.* This work was partially funded by the German Federal Ministry for Economic Affairs and Energy (BMWi) in the scope of the projects CompactWind (FKZ 0325492B) and OWP Control (FKZ 0324131A). The research was partly supported by the German
5 Academic Exchange Service (DAAD) with funds from the Federal Ministry of Education and Research (BMBF). Contributions of the authors affiliated to TU Delft received a partial grant by the European Union's Horizon 2020 research and innovation program under grant agreement No 727477.




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
