# Peer review of "Robust active wake control in consideration of wind direction variability and uncertainty"

_Wind Energy Science, 2018_

## Referee Comment (RC1) · D. P. Held (Referee) · 17 Jul 2018

**Review for "Robust active wake control in consideration of wind direction variability and uncertainty"**

This manuscript investigates how dynamic wind direction changes affect wake deflection wind farm control. The normal distribution is used, which is supported by measurements. Numerical simulations using FLORIS are performed to find optimal yaw angle settings for different wind direction uncertainties, which are later used to find power gains over greedy wind farm control. It is found that the dynamics need to be considered when performing wake steering and failure to do so can lead to power losses compared to greedy optimization.

In general the manuscript is carefully prepared and written in a coherent, understandable way. Figures are explained and discussed clearly. Section 2.2 needs information on how the wind direction was measured. If nacelle anemometry was used a discussion on the effect of the rotor is also necessary. Regarding the bias of wind direction measurement in section 3.1 alternative measurement devices need to be discussed, like nacelle lidars and blade loads. Regarding the controller proposed in section 3 (fig. 10), is the yaw angle changed every iteration? No dead-band is applied? If so, this is not feasible for a utility-scale turbine as was also mentioned in the manuscript earlier.

Minor comments at specific places in the manuscript:

- p.2 ll.20: and missing before 3)
- p.2 ll.30: "A more elaborate…" is not a complete sentence
- p.3 ll.2: hysteresis is not the same as dead-band. Please correct. (Also p.14 ll.12)
- p9. ll.9: Shouldn't ±2sigma cover 95% of the normal distribution?
- p9. ll.31: "In the blue graph (sigma= 4), the yaw misalignment is considerably reduced compared to the first case". This is only true for wakes onto T11 and T13. For the closet turbines there is no reduction in yaw misalignment compared to the first case. I suggest rephrasing this sentence.
- p.11 ll.3: Why was 172deg chosen? Intuitively, I would think that 90 or 270deg gives the largest wake effects.
- Fig. 9: What if you calculate $\int_0^{2\pi} \rho(\varphi) P_{diff} d\varphi$ for the 3 robustness factors? Is this measure positive or negative, i.e. do you get an overall improved performance or not?
- p.19 ll.4: missing bracket
- p.19 ll.13: In the manuscript perfect accuracy (no bias) and imperfect precision (Gaussian error that is introduced in section 3.3) is assumed. Please rephrase
- p.19 ll.14: control instead of contgrol

---

## Short Comment (SC1) · 9 Aug 2018

Overall, this is a great paper and a good contribution to the overall goal to implementing a wind plant wake steering controller.

For figure 2, what instrument was used to measure the wind direction? I think this should be mentioned in the caption and the body of text. It is something to consider when the statistics of the measurement are analyzed, which they are discussed later. If it is a met mast measurement, which way is the boom oriented on the tower? Is tower shadow a concern?

Figure 3: could you add legends to the plots?

In section 3.1, for the stochastic properties of wind direction, I don't see a mention

of atmospheric stability, or turbulence intensity. I would think that these atmospheric quantities would correlate somewhat to the variability of the wind direction. I think it would be good to include something about this in this section.

Overall, it is a great paper!

---

## Author Comment (AC1) · 24 Aug 2018

**Authors reply to comments from referee D.P. Held**

Dear D. P. Held, thank you very much for your thorough review. I will go into the comments in detail below.

**RC1**. *Section 2.2 needs information on how the wind direction was measured.*

**AC**. In section 2.2 a paragraph is added explaining the instrumentation used to measure the inflow:

> The analyses in this article were carried out on the basis of measurements at a met mast. In (Bromm M, et al. Field investigation on the influence of yaw misalignment on the propagation of wind turbine wakes. Wind Energy. 2018;1-18.) measurements from the same device were used. As described there, the height of the met mast is 91.5 m. The wind direction was measured by a wind vane of type 4.3150.00.212 (manufactured by Thies GmbH) at a height of 89.4 m. Additionally, the met mas was equipped with three cup anemometers of type 4.3351.00.000 from the same company, installed at heights of 34.6 m, 89.3 m and 91.5 m. The cup anemometer at 91.5 m was used for filtering the available data as mentioned in Section 2.4 in the manuscript.

**RC2**. *Regarding the bias of wind direction measurement in Section 3.1 alternative measurement devices need to be discussed.*

**AC**. I completely agree that the risk for a considerable bias should be minimized by additional measurement devices. A paragraph is added to the discussion that addresses this point:

> The consideration of the aerodynamic interactions in wind farm control has some critical requirements that must be met as best as possible. This includes the absolute orientation of the wind turbine and a bias in the measurements. While the absolute orientation of the wind turbine is not important for turbine control, it plays a decisive role in wind farm control, as it is required to derive the aerodynamic interactions of the turbines. A bias in wind direction measurement has negative implications for both wind turbine and wind farm control. For this reason, the risk of a significant bias needs to be minimized. Therefore, great care must be taken during installation and alignment of the wind vane. If possible, additional measuring instruments for determining the wind direction should be considered, such as nacelle-mounted lidar or the consideration of blade loads for the determination of the inflow as described in (C.L. Bottasso, C.E.D. Riboldi, Estimation of wind misalignment and vertical shear from blade loads, Renew. Energy. 62 (2014) 293–302).

**RC3**. *Regarding the controller proposed in Section 3 (Fig.10), is the yaw angle changed every iteration? If so, this is not feasible for a utility-scale turbine as was also mentioned in the manuscript earlier.*

**AC**. In Section 3 (Fig.10) the concept of a yaw-controller is introduced. In this concept the yaw-angle of the turbine is adjusted in every iteration, but one iteration takes exactly 5 minutes. Experience from turbine data shows that this is a realistic yaw adjustment rate. This simplified concept of a yaw-controller was used here to be able to validate a large set of data in a manageable fashion.

I fully agree with the referee that constantly adjusting the yaw angle of a utility scale turbine is not feasible. Therefore, the robust yaw control should be carefully integrated into the existing yaw control of the turbine. The reason that only a simplified concept was presented in this research is, that the details of the yaw control of a wind turbine are strongly adapted to the type of wind turbine and are commonly kept confidential.

Minor comments at specific places in the manuscript:

**RC4**. *p.2 ll.20: "and" missing before 3)*

**AC**. Thanks for the comment. It will be corrected.

**RC5**. *p.2 ll.30 "A more elaborate..." is not a complete sentence*

**AC**. The word "For" at the beginning of the sentence is missing and the comma is wrong.

**RC6**. *p.4 ll.2: hysteresis is not the same as dead-band. Please correct. (Also p.14 ll.12)*

**AC**. The wind energy handbook (Burton 2011) broadly describes the yaw controller of a utility scale wind turbine and refers to it as a dead-band controller. The controller uses a threshold before reacting to the deviation of yaw angle and measured wind direction. In literature I did not find that the word hysteresis is clearly defined. In some uses of the term a hysteresis could include a dead-band (e.g. bang-bang control or Schmitt-Trigger) and in

other uses dead-band and hysteresis are clearly distinct, in that at a hysteresis there is always movement and no dead-zone. To be clear in the article I will rephrase the respective paragraphs as follows:

> p.3 ll.19f: The turbulent changes in the wind direction are in contrast to the slowly reacting yaw mechanism of utility-scale wind turbines. The deviation of the wind direction and the yaw angle of the turbine is usually averaged over several minutes and a threshold for the deviation is used to keep the turbine from constantly yawing (Burton, 2011). This has the effect that the turbine is in standstill mode most of the time (Kim and Dalhoff, 2014).

> p.4 ll.2: "Although the details of the yaw control depends on the manufacturer and is commonly kept confidential, in our experience....."

> p.14.ll: "...to simulate additional inaccuracies like for example measurement uncertainties, yaw deviations through the thresholds of the yaw control and alignment errors."

**RC7**. *p.9 ll.9: Shouldn't $\pm 2\sigma$ cover $95\%$ of the normal distribution?*

**AC**. $95\%$ is the commonly used rounded amount, which more accurately the $\pm 1.96\sigma$ region. The more precise $\pm 2\sigma$ region covers $95.45\%$ of the standard normal distribution.

**RC8**. *p.9 ll.31 "In the blue graph ($\sigma = 4°$), the yaw misalignment is considerably reduced compared to the first case". This is only true for wakes onto T11 and T13. For the closest turbines there is no reduction in yaw misalignment compared to the first case. I suggest rephrasing the sentence.*

**AC**. That is correct, the reduction in yaw misalignment for the closest turbines is not well noticeable in the graph. I will rephrase the sentence and address this point

> In the blue graph ($\sigma = 4°$), the yaw misalignment is reduced compared to the first case. This applies in particular for wind directions where the downstream wind turbine is further away (e.g. at around 159° T11 and 201° T13)

**RC9**. *p.11 ll.3: Why was 172° chosen? Intuitively, I would think that 90deg or 270 deg gives the largest wake effects.*

**AC**. It is correct, that the strongest wake effects are at wind directions of 90° and 270°, followed by the wakes at 0° and 180°. There are several reasons why we chose 172° for the exemplary case: First, at 90° and 270° the distance is 3D, although this turbine spacing is sometimes chosen for utility scale turbines, this is only the case for wind directions that occur very rarely. Therefore, this constellation is not very representative. Second, the example at about 180° was used to explain the results of the various robustness parameters and also serves as an example of the passive wake deflection. Third, the particular value of 172° was selected because the results of the optimization for the different robustness parameters show a clear tendency to which direction the wake is deflected and they differ from each other and the baseline.

**RC10**. *Fig. 9: What if you calculate $\int_0^{2\pi} \rho(\varphi) P_{\text{diff}} d\varphi$ for the robustness factors? Is this measure positive of negative, i.e. do you get an overall improved performance of not?.*

**AC**. This is a very good question, which gives the opportunity to explain the results of the optimization in a bit more detail. The optimization is designed to give the best result for a given wind direction distribution, so the optimization with $\sigma = 0°$ should give the highest result for the dirac-delta distribution at 172°. For this simple case we can just look at Fig.9 at the graph at 172°. The black graph shows the highest value of 0.09987 of normalized power, which is slightly higher than the blue graph ($\sigma = 4°$) with 0.09711. The red graph ($\sigma = 8°$) achieves 0.05479. Now, if we weight the depicted graph with a normal distribution with the expected value at 172° and a standard deviation of 4° the results are: black ($\sigma = 0°$) 0.04840, blue ($\sigma = 4°$) 0.05150 and red ($\sigma = 8°$) 0.03419. So for this case the blue graph achieves the highest result, as it was designed for. Finally if we weight the graph with a normal distribution with an expected value of 172° and a standard deviation of 8° the results are: black ($\sigma = 0°$) -0.01841, blue ($\sigma = 4°$) -0.00789 and red ($\sigma = 8°e$) 0.00646. As expected the robust optimization with $\sigma = 8°$ achieves the best result since it was optimized for exactly that case. Furthermore, the results of a certain robustness parameter are always positive for its respective distribution, since the optimization is searching yaw settings that improve the power output compared to the baseline case.

**RC11**. *p.19 ll.4 missing bracket.*

**AC**. Thank you for noticing.

**RC12**.  *p.19 ll.13: In the manuscript perfect accuracy (no bias) and imperfect precision (Gaussian error that is introduced in section 3.3) is assumed. Please rephrase*

**AC**. Thank you for the remark, that in the common technical understanding the word accuracy is synonymous with the word trueness. I will rephrase the sentence as follows

> The method takes dynamic wind direction changes and imprecision in the determination of the wind direction into account within a statistical framework in the optimization.

**RC13**.  *p.19 ll.14 control instead of contgrol.*

**AC**. Of course this was on purpose ;).

---

## Referee Comment (RC2) · Anonymous Referee #2 · 27 Aug 2018

Review of Journal: WES Title: Robust active wake control in consideration of wind direction variability and uncertainty Author(s): Andreas Rott et al. MS No.: wes-2018-50 MS Type: Research articles

The paper deals with wind farm control using yaw stearing and presents a methodology that takes the influence of uncertainty and variability of the wind direction passed to a wake model into account. Among other things it is demonstrated that the application of conventional optimization, where the wind direction is considered as a known input with a sharp value, in some cases may cause energy production to decrease rather than an increase.

The paper is well written, concise and focused, although the line of reasoning is not always clear (see comments below). I recommend the paper for publication with some

revisions and clarifications.

I am missing a discussion of the relation between the readings from a wind vane and the wind direction as a model input. For RANS types of models like FLORIS, the wind direction input is the ensemble mean value. This is NOT a stochastic variable, just a number with a sharp value - no matter how 'dynamic' the wind vane output looks. The Reynolds' decomposition takes care of fluctuations around the mean, so why is it relevant to use a distribution of inputs? You don't offer an explanation. I happen to agree with you that a single five (ten) minute average is insufficicient, although not for reasons you give me. At the same time I don't understand what you are trying to achieve by passing raw wind vane readings as model input.

There IS a statistical uncertainty on five minutes averages (used as estimates of abstract ensemble mean values). However, you study 5 minutes standard deviation (std) estimates of the wind vane data, which is not the same. What is the relevance in terms of model input uncertainty? The std is an upper bound on std of the average, but is it a good one?. You seem to assume that the std is an adequate measure of the robustness. Do you?, and if so why is the std an adequate measure? How should the robustness parameter be quantified?

The control algorithm in sec. 3.3 uses a prediction of the mean wind direction 5 minutes ahead. My guess is that the five minutes separation is a major source of uncertainty. The rms change between consequetive ten minutes averages of the wind direction is typically 5 degrees, a little less for 5 minutes. This could be used as a lower bound on the uncertainty. This can easily be derived from the met data (the pdf is double sided exponential).

Minor things:

'Without loss of generality'. This phrase is used in several places without justification. Please drop it and admit the loss.

[Figure]

figs 11 and 12. Popt/Pgreedy = 0.6% or was it 100.6% ? It is hard to see the differences between 11 and 12. The points melt together so that the density of points is lost. Perhaps smaller point size works better or maybe something different, like error bars?
* * *

---

## Author Comment (AC2) · 17 Sep 2018

**Authors reply to comments from Andrew Scholbrock**

Dear Andrew Scholbrock, thank you very much for your kind remarks and the suggested improvements. In the following I will answer the comments in detail.

**SC1**. *For figure 2, what instrument was used to measure the wind direction? I think this should be mentioned in the caption and the body of text. It is something to consider when the statistics of the measurement are analyzed, which they are discussed later. If it is a met mast measurement, which way is the boom oriented on the tower? Is tower shadow a concern?*

**AC**. The wind direction was measured by a wind vane that was installed on a boom at a met mast in 89.4 m height approx. 1 m above the main structure of the met mast. Small disturbance could be introduced from wind directions around 134° since another boom was located in a distance of 1.65 m in that direction, to which the highest cup anemometer was attached. For the evaluation in the manuscript this was not considered, since this wind direction rarely appeared in the used dataset and the effect should be marginal. I will add the information of this paragraph to section 2.4 of the manuscript.

**SC2**. *Figure 3: could you add legends to the plots?*

**AC**. Legends will be added in the revised version of the manuscript.

**SC3**. *In section 3.1, for the stochastic properties of wind direction, I don't see a mention of atmospheric stability, or turbulence intensity. I would think that these atmospheric quantities would correlate somewhat to the variability of the wind direction. I think it would be good to include something about this in this section.*

**AC**. For the analysis of the stochastic properties of the wind direction only the wind direction measurements were used in the manuscript. However, in the period from 26th July 2016 to the 22nd November 2016 the Monin-Obuhkov length (MOL) could be derived from measurements of a meteorological measuring station at the location. The classification from MOL into stability classes was done according to (Bromm et al., 2018). The histogram of the 5-min standard deviations of the wind directions divided into the stabilities *unstable, neutral* and *stable* is shown in Figure 1 (left). The right figure shows the respective empirical probability for each bin.

[Figure]

Figure 1: (Left) Histogram of 5 minute standard deviations of the wind direction separated in the categories of atmospheric stabilities stable, neutral and unstable. (Right) Normalized histogram of the data

The histogram shows two pronounced maxima. One by approx. 1° and the second by approx. 5.25°. It appears that the distribution consists of two composite distributions. The first, with the focus around 1°, is dominated by measurements in stable atmospheric estimation and the second, with the focus around 5.25°, consists mainly of neutral ones. The higher the standard deviation gets the more likely it is to have unstable stratification, as it can be seen in Figure 1 (right). Figure 1 (left) together with a description will be added to the manuscript to mention the connection of the stochastic properties of the wind direction with the atmospheric stability.

**References**

Bromm, M., Rott, A., Beck, H., Vollmer, L., Steinfeld, G., and Kühn, M.: Field investigation on the influence of yaw misalignment on the propagation of wind turbine wakes, Wind Energy, pp. 1–18, https://doi.org/10.1002/we.2210, URL http://doi.wiley.com/10.1002/we.2210, 2018.

---

## Author Comment (AC3) · 17 Sep 2018

**Authors reply to comments from referee #2**

Dear Referee # 2
thank you very much for the review and the comments, which I will discuss in detail below.

**RC1**. *I am missing a discussion of the relation between the readings from a wind vane and the wind direction as a model input. For RANS types of models like FLORIS, the wind direction input is the ensemble mean value. This is NOT a stochastic variable, just a number with a sharp value - no matter how 'dynamic' the wind vane output looks. The Reynolds' decomposition takes care of fluctuations around the mean, so why is it relevant to use a distribution of inputs? You don't offer an explanation. I happen to agree with you that a single five (ten) minute average is insufficient, although not for reasons you give me. At the same time I don't understand what you are trying to achieve by passing raw wind vane readings as model input*

**AC**. The wake model FLORIS, which was used for this investigation, is a combination of the Jensen model (Jensen, 1983; Katic et al., 1986) with the wake deflection model presented by Jiménez (Jiménez et al., 2009) and further adjustments, detailed in (Gebraad et al., 2016). The model is parameterized to match power measurements obtained from 10-min LES simulations. The input of the model is the main wind direction, which, I agree, also represents the empirical mean wind direction. However, as already mentioned in the manuscript, conventional LES simulations do not reproduce dynamic changes of the wind direction, but rather consider a constant main wind direction. Only comparably small fluctuations of the wind direction are generated by the simulated turbulence. While this is useful for the qualitative analysis of specific situations, it does not represent the measurements we have made in the field. Gaumond et al. (Gaumond et al., 2014) proposed to use a wide span of wind directions as model input and to calculate the (normally distributed) weighted average to successfully account for the measured distribution of wind directions within 10 minutes. Among others he also used the Jensen model. We have extended this approach to also account for other uncertainties, as explained in the manuscript, and applied it to 5 minute periods of time within an optimization. The results of Gaumond are also the motivation to use the empirical 5-min dstribution of wind vane measurements as input for the model in the evaluation of control algorithms in Section 3.3. in the manuscript.

For a better explanation of the method in the manuscript Section 3 was revised and the following paragraph is added:

> p.9 ll. 8: This is in accordance with (Gaumond et al., 2014), who used a range of wind directions together with weightings corresponding to normal distributions as model input for similar wake models to take into account the variability of 10 min wind direction time series, and thus could improve the agreement of model results with measurements.

> p.14. ll. 18: Also, according to (Gaumond et al., 2014) the wind direction distribution as model input gives a better agreement with measured data. Therefore we use the empiric distribution for every time series, respectively.

**RC2**. *There IS a statistical uncertainty on five minutes averages (used as estimates of abstract ensemble mean values). However, you study 5 minutes standard deviation (std) estimates of the wind vane data, which is not the same. What is the relevance in terms of model input uncertainty? The std is an upper bound on std of the average, but is it a good one?. You seem to assume that the std is an adequate measure of the robustness. Do you?, and if so why is the std an adequate measure? How should the robustness parameter be quantified?*

**AC**. In the manuscript the hypothesis was tested whether the 5 minutes wind direction time series can be approximated by a normal distribution. Since this hypothesis was confirmed for the majority of the data, the standard deviations of the time series were calculated because the statistics of normally distributed random variables can be fully described by the mean and the standard deviation. The standard deviation therefore serves only to describe the respective time series and should not be used to estimate the uncertainty of the mean. In order to consider the wind direction changes within the 5 minutes in the optimization, a normally distributed weighting was used in the optimization according to Gaumond (Gaumond et al., 2014). In the evaluation, it was examined whether this method can also be used to compensate for further uncertainties, such as measurement inaccuracies and the deviation due to the time offset when the yaw angle follows the wind direction measurement. Increasing the robustness parameter and thus the standard deviation in the optimization implies that all errors are assumed to be normally distributed, since only in this case can the variances of the various uncertainties be summed up. We have not tested this hypothesis for the other uncertainties, but nevertheless the results show that the normal distribution is sufficient to improve the robustness of the control. To clarify this issue in the manuscript a paragraph will be added, which simultaneously addresses the next comment (see below) As already mentioned in the discussion of the manuscript, the results should only be understood as an indication and further investigations with simulations of higher fidelity and measurements in the free field should be carried out.

**RC3**. *The control algorithm in sec. 3.3 uses a prediction of the mean wind direction 5 minutes ahead. My guess is that the five minutes separation is a major source of uncertainty. The rms change between consequetive ten minutes averages of the wind direction is typically 5 degrees, a little less for 5 minutes. This could be used as a lower bound on the uncertainty. This can easily be derived from the met data (the pdf is double sided exponential).*

**AC**. The prediction of the mean wind direction of the 5 minutes ahead is based on the persistence method, which means that the control algorithm assumes that the best guess for the mean of the next 5 minutes wind direction time series is the mean of the current time series. The deviation of the mean is treated as uncertainty and I agree that this causes a large part of the overall uncertainty. As mentioned above, we assume in the manuscript that the robustness parameter and thus the normally distributed wind direction input of the optimization compensates this uncertainty accordingly. However, it was not tested whether this error is normally distributed. Therefore, we analysed the deviations of the successive mean values and created the histogram see Figure 1. As you mentioned the double sided exponential (Laplace) distribution can be fitted reasonably well to the empirical probability density distribution, although in this case a location scaled t-distribution results in a better fit (see Figure 1 (right)).

[Figure]

Figure 1: (Left) Histogram of the changes of the mean values of successive wind direction time series with a fitted Laplace distribution. (Right) Histogram of the changes of the mean values of successive wind direction time series with a fitted t-distribution.

The optimization is designed to be able to incorporate any kind of distribution as a component and therefore it is possible to integrate a t-distribution (or other distribution) in combination with a normal distribution by the convolution of both, however we do not change the presented method in the manuscript for two reasons. First, we want to demonstrate a rather simple method for the robust control algorithm and the single robustness parameter makes it easy and intuitive to adjust the robustness. Second, the results in the evaluation of the robust control algorithm show promising results with the current method.

To improve the manuscript in terms of clarity the right part of Figure 1 and the following paragraph will be added to the discussion part:

> p.18 ll.13f: In addition, we have assumed that all uncertainties that occur can be estimated by a normal distribution. This assumption proved to be sufficient in the evaluation, but individual sources of uncertainty can still be further investigated. The deviations of the mean values from successive wind direction time series, for example, seem to be well described by a t-distribution (see Figure 1). This finding could be used for the selection of the weightings in the optimization to improve the optimization. We have decided against this at this point, since an important aspect of this yaw control is its relative simplicity. By integrating a further distribution (by convolution with the normal distribution) the robustness could no longer be described by the robustness parameter alone.

**RC4**. *'Without loss of generality'. This phrase is used in several places without justification. Please drop it and admit the loss.*

**AC**. Thank you for this remark. The places where this was used were revised and adjusted as follows:

> p.5 ll.15f: We assume that the turbines run at a constant axial induction factor of $a_j = \frac{1}{3}$ for all $j$ and the power output $P_j$ of each turbine is normalized with respect to the power output of a turbine in undisturbed inflow conditions, since we are focussing on the influence of the yaw angle on the relative change of turbine power.

> p.9 ll.15: For the analysis we assume that a wind direction bias is negligible.

**RC5**. *figs 11 and 12. Popt/Pgreedy = 0.6between 11 and 12. The points melt together so that the density of points is lost. Perhaps smaller point size works better or maybe something different, like error bars?*

**AC**. For better understandability the variable $P_{\text{gain}} := \overline{\frac{P_{\text{opt}}}{P_{\text{greedy}}}} - 1$ is introduced in Section 3.3. and for better visibility of the scatter density in the Figures 11 and 12 the upper and lower quartiles as well as the median are added to the plots.

**References**

Gaumond, M., Réthoré, P. E., Ott, S., Peña, A., Bechmann, A., and Hansen, K. S.: Evaluation of the wind direction uncertainty and its impact on wake modeling at the Horns Rev offshore wind farm, Wind Energy, 17, 1169–1178, https://doi.org/10.1002/we.1625, URL `http://onlinelibrary.wiley.com/doi/10.1002/we.1608/full`, 2014.

Gebraad, P. M., Teeuwisse, F. W., Van Wingerden, J. W., Fleming, P. A., Ruben, S. D., Marden, J. R., and Pao, L. Y.: Wind plant power optimization through yaw control using a parametric model for wake effects - A CFD simulation study, Wind Energy, 19, 95–114, https://doi.org/10.1002/we.1822, URL `http://onlinelibrary.wiley.com/doi/10.1002/we.1608/full`, 2016.

Jensen, N. O.: A note on wind generator interaction, Risø-M-2411 Risø National Laboratory Roskilde, pp. 1–16, https://doi.org/Riso-M-2411, URL `http://www.risoe.dk/rispubl/VEA/veapdf/ris-m-2411.pdf`, 1983.

Jiménez, Á., Crespo, A., and Migoya, E.: Application of a LES technique to characterize the wake deflection of a wind turbine in yaw, Wind Energy, 13, 559–572, https://doi.org/10.1002/we.380, URL `http://onlinelibrary.wiley.com/doi/10.1002/we.380/full` `http://doi.wiley.com/10.1002/we.380`, 2009.

Katic, I., Højstrup, J., and Jensen, N.: A simple model for cluster efficiency, European wind energy association conference and exhibition, pp. 407–410, URL `http://forskningsbasen.deff.dk/Share.external?sp=S3a811668-6814-4671-af17-0595fd17b8f6sp=Sdtu`, 1986.

---

## Author Response (AR1)

The author's responses are given in detail in the author's comments AC1, AC2 and AC3. The according adaptations of the manuscript are highlighted in boxes. Furthermore in the following the latexdiff between the first submission and the revised version is given.

[revised manuscript text omitted]